# Development of a Gene Delivery System of Oligonucleotides for Fibroses by Targeting Cell-Surface Vimentin-Expressing Cells with N-Acetylglucosamine-Bearing Polymer-Conjugated Polyethyleneimine

**DOI:** 10.3390/polym12071508

**Published:** 2020-07-07

**Authors:** Inu Song, Hirohiko Ise

**Affiliations:** 1Graduate School of Engineering, Kyushu University, 744 Motooka Nishi-ku, Fukuoka 819-0395, Japan; inu.song@ms.ifoc.kyushu-u.ac.jp; 2Institute for Materials Chemistry and Engineering, Kyushu University, 744 Motooka Nishi-ku, Fukuoka 819-0395, Japan

**Keywords:** fibrosis, myofibroblasts, activated stellate cells, vimentin, NF-κB decoy oligonucleotides, HSP47, siRNA

## Abstract

Targeting myofibroblasts and activated stellate cells in lesion sites of fibrotic tissues is an important approach to treat fibroses. Herein, we focused on targeting the cytoskeletal proteins vimentin, which are reportedly highly expressed on the surface of these cells and have N-acetylglucosamine (GlcNAc)-binding activity. A GlcNAc-bearing polymer synthesized via radical polymerization with a reversible addition-fragmentation chain transfer reagent has been previously found to interact with cell-surface vimentin-expressing cells. We designed a GlcNAc-bearing polymer-conjugated polyethyleneimine (PEI), as the gene carrier to target cell-surface vimentin-expressing cells and specifically deliver nuclear factor-κB decoy oligonucleotides (ODNs) and heat shock protein 47 (HSP47)-small interfering RNA (siRNA) to normal human dermal fibroblasts (NHDFs) that express cell-surface vimentin. The results showed that the expression of tumor necrosis factor-α in lipopolysaccharide-stimulated NHDFs and HSP47 in transforming growth factor-β1-stimulated NHDFs was suppressed by cellular uptake of the GlcNAc-bearing polymer-conjugated PEI/nuclear factor (NF)-κB decoy ODNs and HSP47-siRNA complexes through cell-surface vimentin, respectively. These findings suggest that the effective and specific delivery of ODNs and siRNA for cell-surface vimentin-expressing cells such as myofibroblasts and activated stellate cells can be achieved using GlcNAc-bearing polymer-conjugated PEI. This therapeutic approach could prove advantageous to prevent the promotion of various fibroses.

## 1. Introduction

Various tissue fibroses such as liver cirrhosis, pulmonary fibrosis, and hypertensive heart disease are caused by chronic inflammation related to repeated tissue injuries and autoimmune diseases [1]. When conventional tissue-repairing systems are disrupted, chronic inflammation is induced and many inflammatory cytokines, chemokines, and growth factors are secreted from immune cells such as leukocytes and macrophages. These cytokines and growth factors stimulate the activation of fibroblasts and stellate cells, inducing their transformation into myofibroblasts and activated stellate cells, respectively [1,2,3]. These cells then produce transforming growth factor-β (TGF-β), which induces their proliferation and the production of extracellular matrix (ECM) constituents such as collagen [4,5]. The accumulation of excess ECM leads to the stiffening of parenchymal tissues, which in turn leads to the destruction of their structural architecture and function [6]. These fibrotic processes are common to various tissues and organs, and the large numbers of myofibroblasts and activated stellate cells accompanied by abnormal ECM deposition exacerbate fibroses [1,7,8,9,10,11]. In tumors, desmoplastic phenomena and ECM deposition are induced by cancer-associated fibroblasts like myofibroblasts, which are one of the most crucial cells for tumor progression, leading to fibrotic reactions similar to organ fibroses [12]. The accurate and early diagnosis of these fibroses is important for tumor prevention and to monitor progression [13].

To aid the early diagnosis of various fibroses, in this study we focused on the emergence of myofibroblasts and activated stellate cells in fibrotic tissues. We predicted that the detection of the appearance of these cells in the early stages of fibroses would provide a useful diagnostic tool. To this end, the cell surface cytoskeletal proteins vimentin and desmin were used as markers to target myofibroblasts and activated hepatic stellate cells. Namely, the type III intermediate filament proteins vimentin, desmin, glial fibrillary acidic protein, and peripherin are expressed on the surface of various cells and possess N-acetylglucosamine (GlcNAc)-binding activity via GlcNAc-bearing polymers [14,15,16,17,18,19]. Moreover, vimentin and desmin are highly expressed on myofibroblasts and activated stellate cells in various lesion sites [20,21], and it is possible to target activated stellate cells in liver fibrosis with bio-imaging systems that utilize GlcNAc-conjugated polyethyleneimine (GlcNAc-PEI) and small interfering (siRNA) delivery systems [22,23]. Namely, GlcNAc-PEI accumulates in fibrotic but not normal liver tissues, and it specifically interacts with activated stellate cells that express desmin, suggesting that desmin and vimentin can be highly expressed on the cell surface of myofibroblasts and activated stellate cells in fibrotic tissues. Therefore, we predicted a therapeutic and early diagnostic approach for fibroses based on the interaction of cell surface-expressing desmin and vimentin with GlcNAc-conjugated materials.

To date, GlcNAc-PEIs have been produced by randomly conjugating GlcNAc to PEI [22,23,24], thus making it difficult to reproduce and precisely control the degree of GlcNAc-conjugation. Therefore, it is important to develop GlcNAc-bearing polymers with high affinity to cell surface vimentin and desmin, and the production of gene carriers that are conjugated to these GlcNAc-bearing polymers might be advantageous to target myofibroblasts and activated stellate cells. We previously succeeded in producing a well-defined and highly hydrophilic GlcNAc-bearing polymer, polyacrylate-GlcNAc (AC-GlcNAc), which has high affinity to cell surface-expressed type-III intermediate filament proteins [14,18]. It was composed of carboxyethyl acrylate as the main chain and GlcNAc as the side chain and was produced through polymerization with reversible addition-fragmentation chain transfer (RAFT) reagents. It has been reported that cell surface vimentin and desmin can interact with multivalent GlcNAc-bearing polymers but not GlcNAc monosaccharides [15]. To design gene carriers to effectively target the lesion sites of fibrotic tissue while avoiding the risk of vascular embolization, it is important to produce hydrophilic gene carriers that are as small as possible. We previously discovered the optimal valency of GlcNAc-bearing polymers for binding to cell surface vimentin by using various sized AC-GlcNAcs produced with reversible deactivation radical polymerization (RDRP). The optimal valency of AC-GlcNAc was determined to be approximately 10 GlcNAc ligands [14,17]. AC-GlcNAc produced from approximately 10 GlcNAc ligands and an average molecular weight (M_w_) of approximately 4 kDa was found to have high affinity for vimentin and desmin (K_D_: 3 × 10^−8^ M and 2 × 10^−8^ M, respectively). RDRP achieves the production of various size-regulated AC-GlcNAc. Moreover, some functional groups such as carboxyl and thiol groups can be added to the terminal regions of AC-GlcNAc via RDRP. In the present study, we tried to make the smallest-sized and well-defined AC-GlcNAc to which the thiol group is introduced for conjugation with PEI via polymerization with RAFT reagents, and to design gene carriers to target cell surface vimentin and desmin by conjugating AC-GlcNAc to PEI via disulfide bonds. AC-GlcNAc has a dodecylthiocarbonothioylthio group in the terminal region, which can be removed by reducing conditions, exposing the SH group in the terminal region. In contrast, branched PEI can be conjugated with *N,N*′-bis(acryloyl)cystamine (CBA) through a Michael addition, a process in which some SH groups of the branched PEI are engaged [25]. Therefore, AC-GlcNAc-conjugated branched PEI was produced via disulfide bond formation.

We focused on nuclear factor (NF)-κB decoy oligonucleotides (ODNs) as anti-inflammatory factors delivered to vimentin- and desmin-expressing cells in fibrotic lesion sites. NF-κB decoy ODNs encode a sequence for an NF-κB transcription factor binding site and can bind NF-κB transcription factors in the cytoplasm, inhibiting their import into the nucleus [26,27,28]. Since suppression of the transcriptional activities of NF-κB decreases the production of various inflammatory cytokines, the specific delivery of NF-κB decoy ODNs to fibrotic lesions is expected to yield anti-inflammatory effects and suppress the fibrotic progression. Previously, we reported that GlcNAc-bearing polymer-coated liposomes containing NF-κB decoy ODNs were taken up into vascular smooth muscle cells (VSMCs) and suppressed expression of the inflammatory cytokine interleukin-6 (IL-6) in IL-1β-stimulated VSMCs [29]. Moreover, we demonstrated the targeting of VSMCs in injured blood vessels by GlcNAc-bearing polymer-coated liposomes [29]. Therefore, although it is difficult to adequately incorporate NF-κB decoy ODNs into these liposomes, NF-κB decoy ODNs are expected to effectively suppress inflammation in inflammatory lesions. Based on these findings, we aimed to deliver NF-κB decoy ODNs to myofibroblasts specifically and effectively by targeting surface-expressing vimentin and desmin with AC-GlcNAc-conjugated branched PEI. Next, we focused on the delivery of heat shock protein 47 (HSP47)-siRNA to vimentin- and desmin-expressing cells in fibrotic lesion sites. As previously mentioned, since myofibroblasts and activated stellate cells abundantly produce collagen, promoting a fibrotic response, the inhibition of collagen production by these cells presents an effective way to improve the symptoms of various fibroses. Collagen production by these cells is regulated by HSP47 and lysyl hydroxylase [30,31,32]. TGF-β activates myofibroblasts and activated stellate cells and promotes collagen production [33,34,35]. Especially, many therapeutic approaches for fibroses by suppressing HSP47 with the delivery of HSP47-siRNA to fibrotic tissues have been previously reported [36,37,38]. Therefore, the delivery of siRNA for HSP47, lysyl hydroxylase, and TGF-β to myofibroblasts and activated stellate cells in fibrotic lesion sites with our strategy represents a useful therapeutic approach.

To specifically and effectively target cell-surface vimentin of myofibroblasts and activated stellate cells, nonspecific interactions induced by the cationic character of PEI with cells must be suppressed. AC-GlcNAc-conjugated branched PEI that can interact with HeLa cells, cervical cancer cells, but not with vimentin-deficient HeLa cells is presumed to be optimal for the specific recognition of cell-surface vimentin. We determined the optimal conjugation ratio of branched PEI to AC-GlcNAc by altering the amount of PEI conjugated to AC-GlcNAc using HeLa cells and vimentin-deficient HeLa cells. Moreover, normal human dermal fibroblasts (NHDFs) were used as myofibroblasts and activated stellate cells in this study. NHDFs that are cultured long-term express α-smooth muscle actin and have myofibroblast-like characteristics. To develop a gene carrier to target lesion sites of fibrotic tissues, the interaction and the anti-inflammatory effect of NHDFs with AC-GlcNAc-conjugated PEI/NF-κB decoy ODN complexes were examined. Moreover, to test the suppression of fibrotic progression, we examined whether the expression of HSP47 in TGF-β-stimulated NHDFs could be suppressed by the delivery of HSP47-siRNA with AC-GlcNAc-conjugated branched PEI.

## 2. Materials and Methods

### 2.1. Materials

Branched 2-kDa PEI and CBA were purchased from Polyscience (Warrington, PA, USA). 2,2′-Azobis(isobutyronitrile) (AIBN) was purchased from Tokyo Chemical Industry Co. (Tokyo, Japan). 2-(Dodecylthiocarbonothioylthio)-2-methylpropanoic acid (DTMPA) and mouse monoclonal anti-β-actin antibody were purchased from Sigma-Aldrich (St. Louis, MO, USA). Dulbecco’s modified Eagle’s medium (DMEM, High glucose), Accutase™, and protease and phosphatase inhibitor cocktail were purchased from Nacalai Tesque (Kyoto, Japan). Lipopolysaccharide (LPS) from *Escherichia coli* O55 (retrieved by phenol extraction) was purchased from Fujifilm (Tokyo, Japan). TGF-β1 was purchased from Cell Guidance Systems LLC (St. Louis, MO, USA). Human fibroblast defined medium (HFDM)-1 was purchased from Cell Science and Technology Institute (Miyagi, Japan). Rabbit polyclonal anti-tumor necrosis factor (TNF)-α antibody and rabbit polyclonal anti-HSP47 antibody were purchased from Proteintech (Rosemont, IL, USA). Mouse monoclonal anti-β-actin antibody was purchased from Sigma-Aldrich. Horseradish peroxidase (HRP)-conjugated anti-rabbit and anti-mouse IgG antibodies were purchase from Jackson ImmunoResearch (West Grove, PA, USA). NF-κB decoy ODNs (sequences: carboxyfluorescein (FAM)-labeled 5′-CCT TGA AGG GAT TTC CCT CC-3′ and 3′-GGA GGG AAA TCC CTT CAA GG-5′) were ordered from Thermo Fisher Scientific (Waltham, MA, USA). HSP47-siRNA (sequence: 5′-AGC CCU CUU CUG ACA CUA Att-3′ and 3′-UUA GUG UCA GAA GAG GGC Ugg-5′) and Silencer™ Select Negative Control No. 1 siRNA (Negative control-siRNA) were ordered as Silencer^®^ Select Pre-designed siRNA from Thermo Fisher Scientific (Waltham, MA, USA).

### 2.2. Polymer Preparation and AC-GlcNAc-conjugated PEI Characterization

The method for synthesizing the GlcNAc-bearing polymer AC-GlcNAc has been described in our previous report [14]. Briefly, AC-GlcNAc monomers were prepared by conjugating carboxyethyl acrylate to GlcNAc-NH_2_. GlcNAc-NH_2_ was prepared by aminating position 1 of GlcNAc with a saturated NH_4_HCO_3_ solution [14]. These monomers were then polymerized by the initiator AIBN and the RAFT agent DTMPA at a molar ratio of 1:10 of AC-GlcNAc monomers to DTMPA. The molecular weight of AC-GlcNAc was measured by gel permeation chromatography (GPC; LC-9110G NEXT with the JAIGEL-GS510 Column; Japan Analytical Industry Co., Tokyo, Japan) with 200 mM NaNO_3_/20% acetonitrile in water. The molecular weight of AC-GlcNAc was determined with a calibration curve based on a pullulan standard (Appendix A). Disulfide-containing PEI was obtained via Michael addition between 2-kDa branched PEI and CBA [22,25]. Briefly, PEI (in 10% aqueous methanol) and CBA (at a molar ratio of 1:1.6 of the reactive group of CBA to the primary amine residue of PEI) were mixed and stirred under a nitrogen atmosphere for 24 h, at 45 °C. Subsequently, excess PEI was added to eliminate unreacted CBA. After that, the products were purified using a dialysis membrane (molecular weight cut-off: 3500) for 2 days. AC-GlcNAc-conjugated PEIs were synthesized through a thiol-disulfide exchange reaction. To perform the thiol-disulfide exchange reaction, AC-GlcNAc was reduced by NaBH_4_, and the reduced AC-GlcNAc (AC-GlcNAc-SH) was then neutralized and reacted with CBA-conjugated PEI. First, 10 mg of AC-GlcNAc and 5 mg of NaBH_4_ were mixed in 200 µL of distilled water and incubated for 1 h at room temperature (RT). After incubating, the reduced AC-GlcNAc (AC-GlcNAc-SH) was precipitated with 1.5 mL of 2-propanol. AC-GlcNAc-SH was dissolved in 1 mL of distilled water and neutralized with 1 M sodium acetate buffer (pH 4), and then AC-GlcNAc-SH and CBA-conjugated PEI were mixed at RT under atmospheric conditions overnight. Next, the reaction mixtures were dialyzed for 1 day to remove unreacted AC-GlcNAc-SH using a dialysis membrane (molecular weight cut-off: ~8 kDa). Lastly, the products were obtained through lyophilization for 1 day. Since disulfide bonds between AC-GlcNAc and CBA-conjugated PEI were systematically generated, these polymers could achieve various cationic properties by altering the molar ratio of AC-GlcNAc to PEI (2 kDa). In this study, we prepared various types of AC-GlcNAc-conjugated PEIs by mixing AC-GlcNAc-SH and CBA-conjugated PEI at molar ratios of 1, 5, 10, and 20. To confirm the syntheses of CBA-conjugated PEI and AC-GlcNAc-conjugated PEI, ^1^H NMR of AC-GlcNAc, CBA-conjugated PEI, and AC-GlcNAc-conjugated PEI were measured with the Advance III HD 400 MHz (Bruker, Ettlingen, Germany) with DMSO-d_6_ containing 0.03 wt. % tetramethyl silane and D_2_O containing 0.05 wt. % 3-(trimethylsilyl)propionic-2,2,3,3-d_4_ acid, sodium salt (Sigma-Aldrich) as the solvents, respectively.

### 2.3. Preparation and Characterization of the AC-GlcNAc-Conjugated PEI and NF-κB Decoy ODN Complexes

To prepare adequately condensed complexes, 2.4 µg of NF-κB decoy ODNs and AC-GlcNAc-conjugated PEI were mixed at various weight ratios (*w/w*) ranging from 1 to 40 in 150 mM NaCl and incubated for 30 min at 37 °C. To confirm the formation of complexes, a gel retardation assay was performed. These complexes were electrophoresed on a 2% (*w/w*) agarose gel and the mobility shifts of these complexes were observed. The size and zeta potential of these complexes were measured by dynamic light scattering (DLS) with a Zetasizer nano (Malvern Panalytical, Malvern, UK).

### 2.4. Interaction of Complexes with Cell Surface Vimentin

The interactions of AC-GlcNAc-conjugated PEI/FAM-labeled NF-κB decoy complexes with HeLa cells, vimentin-deficient HeLa cells, and NHDFs were analyzed by flow cytometry (Guava^®^ easyCyte™, EMD Millipore, Billerica, MA, USA) and fluorescent microscopy (BZ-X700, Keyence, Osaka, Japan). The complexes including 2.4 µg of NF-κB decoy ODNs were prepared in serum-free DMEM for 30 min at 37 °C. HeLa cells, vimentin-deficient HeLa cells, and NHDFs were seeded onto 35-mm dishes at a density of 1 × 10^4^, 1 × 10^4^, and 2 × 10^4^ cells/dish, respectively. Vimentin-deficient HeLa cells were produced by transfection with a human vimentin CRISPR/Cas9-knockout plasmid and human vimentin homology-directed DNA repair plasmid (Santa Cruz Biotechnology, Dallas, TX, USA), as described previously [14]. NHDFs were purchased from TAKARA BIO (Shiga, Japan), and cultured in DMEM containing 10% fetal bovine serum (FBS) for 3 days at 37 °C in a 95% air/5% CO_2_ atmosphere. Only vimentin-deficient HeLa cells were added to 40 µg/mL puromycin (Nacalai Tesque, Kyoto, Japan) to maintain the vimentin deficiency condition. After 3 days of culture, the complexes were added to the cultured cells and incubated for 1 h. The unreacted complexes were removed by washing with PBS three times. The cultured cells were further incubated for 2 h with DMEM containing 10% FBS. The cells were then fixed in 4% paraformaldehyde/PBS for 30 min at RT. The nuclei were stained with a 4′,6-diamidino-2-phenylindole solution. The interaction between AC-GlcNAc-conjugated PEI/FAM-labeled NF-κB ODN complexes and the cultured cells was observed by fluorescence microscopy with a BZ-X700 instrument (Keyence, Osaka, Japan). Flow cytometric analysis of the interaction of the complexes with the cultured cells was performed using the Guava^®^ easyCyte™ flow cytometer (EMD Millipore). Briefly, the cultured cells were detached with Accutase™ and then suspended in 1 mL of PBS. The cell suspensions were then analyzed by flow cytometry.

### 2.5. Western Blotting to Detect TNF-α and HSP4 Expressed by LPS- and TGF-β1-stimulated NHDFs, Respectively

To investigate the anti-inflammatory effect and the expression of HSP47 of incorporating each complex into NHDFs, the secreted level of the pro-inflammatory cytokine TNF-α and the expression of HSP47 were analyzed by Western blotting, respectively. NHDFs were cultured in 35-mm dishes in DMEM containing 10% FBS at a cell density of 5 × 10^4^ cells/dish for 1 day. After 1 day of culture, the media were replaced with serum-free HFDM-1 containing 10 ng/mL epidermal growth factor and cultured for 1 day. In case of detection of TNF-α, after 1 day of incubation, the expression of TNF-α by NHDFs was stimulated with 5 µg/mL LPS. Simultaneously, the complexes (*w/w* ratio of 10) including 2.4 µg of NF-κB decoy ODNs were added to the cultured NHDFs. Next, at 8 h after the addition, the conditioned media and total cell lysate were collected from these cultured NHDFs. In case of detection of HSP47, the siRNA-complex was prepared by incubating the mixture of 12 µg AC-GlcNAc-conjugated PEI (1:10) and 1.2 µg HSP47- or Negative control-siRNA (*w/w* ratio of 10) in serum-free HFDM-1 at 37 °C for 30 min. Then, 10 ng/mL TGF-β1 and each siRNA complex were added to the cultured NHDFs. At 48 h after the addition, total cell lysate was collected from these cultured NHDFs. Briefly, the cells were lysed in lysis buffer (1% triton-X 100, 20 mM HEPES-NaOH, 500 mM NaCl, protease, and phosphatase inhibitor cocktail), and the lysates were then purified by centrifugation at 20,000× *g* for 15 min at 4 °C. The lysate and the conditioned media were analyzed by SDS-PAGE, and the separated proteins were transferred onto a PVDF membrane (Millipore). Subsequently, the membrane was blocked with Blocking One (Nacalai Tesque) for 30 min and incubated with rabbit polyclonal anti-TNF-α antibody, rabbit polyclonal anti-HSP47 antibody, or mouse monoclonal anti-β-actin antibody as the primary antibodies at 4 °C overnight. HRP-conjugated anti-rabbit or mouse antibodies were added as secondary antibodies and incubated for 1 h at RT. Detection was performed using a C-DiGit^®^ Blot Scanner (LI-COR, Lincoln, NE, USA), and the relative band intensities were normalized through densitometric analysis using Image Studio Lite software (LI-COR).

### 2.6. Cell Viability of NHDFs with AC-GlcNAc-conjugated PEI Complexes

To evaluate the cytotoxic effects of AC-GlcNAc-conjugated PEI complexes on NHDFs, we performed a Cell Counting Kit-8 (CCK-8) assay (DOJINDO LABORATORIES, Kumamoto, Japan). AC-GlcNAc-conjugated PEI complexes (1:1, 1:5, 1:10, and 1:20) including 2.4 µg NF-κB decoy ODNs were cultured with NHDFs for 1 day. After 1 day of culture, CCK-8 solution was added to these treated NHDFs at a final concentration of 10% and then NHDFs were incubated for 1 h. After 1 h of incubation, the absorbance (450 nm) of these conditioned media was measured with a microplate reader (Bio-Rad Laboratories, Inc., Hercules, CA, USA).

### 2.7. Statistical Analysis

Data are described as mean values ± standard deviations (SDs) of more than three independent experiments. Significant differences between two groups were estimated by unpaired Student’s *t*-tests using Microsoft Excel for Mac (ver. 16.34). *p*-values < 0.01 were considered as a significant difference between the designated groups.

## 3. Results

### 3.1. Preparation of AC-GlcNAc-conjugated PEI

The conjugation of branched PEI to CBA was confirmed by proton nuclear magnetic resonance (^1^H NMR; Scheme 1, Figure 1). -SCH_2_- proton peaks of CBA appeared at δ = 2.4 ppm, and -CH_2_CH_2_NH- proton peaks of PEI appeared between δ = 2.6 ppm and δ = 3.1 ppm. Moreover, the acrylamide residue peaks of CBA at δ = 5.8 and 6.5 ppm disappeared, indicating that the conjugation of PEI to CBA was carried out successfully. The molar ratio of CBA introduced to branched PEI was approximately 6.9 mol per 1 mol of PEI, according to the integral ratio of proton peaks that correspond to CBA and PEI in the NMR spectrum. The GPC chromatogram indicated that AC-GlcNAc was produced with a M_w_ of 4.7 kDa (GlcNAc ligands, 13) and a polymer dispersity index of 1.4 through polymerization with RAFT (Appendix A). These results demonstrated that well-defined AC-GlcNAc was produced by molecular size-regulation with RDRP.

AC-GlcNAc-SH was crosslinked with CBA-conjugated PEI under oxidative conditions and the crosslinking was confirmed by ^1^H NMR. -COCH_3_- proton peaks of AC-GlcNAc appeared at δ = 1.9 ppm and the proton peaks (between δ = 2.6 ppm and δ = 3.1 ppm) of -CH_2_CH_2_NH- residues of CBA-conjugated PEI were detected, indicating that AC-GlcNAc-SH was successfully crosslinked to branched PEI (Scheme 2). ^1^H NMR analysis revealed the number of PEI per AC-GlcNAc (Table 1, Figure 2, Appendix A).

### 3.2. Characterization of the AC-GlcNAc-conjugated PEI and NF-κB Decoy Oligonucleotide Complex

The formation of complexes of each AC-GlcNAc-conjugated PEI and NF-κB decoy ODN was confirmed by a gel retardation assay. The complete condensation of the complexes of AC-GlcNAc-conjugated PEI and NF-κB decoy ODNs at ratios of 1:1, 1:5, 1:10, and 1:20 was observed in AC-GlcNAc-conjugated PEI/NF-κB decoy ODNs at *w/w* ratios of 30, 12, 10, and 10, respectively (Figure 3). Next, DLS analysis demonstrated that the sizes of these complexes decreased as the conjugation ratio of PEI to AC-GlcNAc-conjugated PEI increased (Figure 4a). The zeta potential of these complexes also increased as the conjugation ratio of PEI increased (Figure 4b). These results indicate that an increase in the conjugation ratio of PEI to AC-GlcNAc elevates the cationic characteristic and condensation of the complexes for ODNs (Table 2).

### 3.3. Evaluation of the Interaction between AC-GlcNAc-conjugated PEI and NF-κB Decoy ODN Complexes and Cell-Surface Vimentin

The interactions between AC-GlcNAc-conjugated polyethyleneimine (PEI) (1:1, 1:5, 1:10, and 1:20) and NF-κB decoy oligonucleotide complexes (*w/w* ratios of 30, 12, 10, and 10, respectively) and vimentin-expressing cells were examined by flow cytometry with HeLa cells, vimentin-deficient HeLa cells, and NHDFs. To determine the optimal AC-GlcNAc-conjugated PEI complex that can specifically interact with vimentin-expressing cells such as NHDFs though cell surface vimentin rather than the cationic characteristic of PEI, the specific interaction of these complexes was examined by comparing HeLa cells with vimentin-deficient HeLa cells. Moreover, the interaction between these complexes and NHDFs as myofibroblasts and activated stellate cells was examined. The results showed that the interaction between the AC-GlcNAc-conjugated PEI complex (1:1) and HeLa cells was not as strong as that with vimentin-deficient HeLa cells (Figure 5). The interaction between the AC-GlcNAc-conjugated PEI complex (1:5 and 1:10) and HeLa cells was strong, and these complexes did not interact with vimentin-deficient HeLa cells (Figure 5). Moreover, the AC-GlcNAc-conjugated PEI complex (1:5 and 1:10) was demonstrated to also strongly interact with NHDFs. The AC-GlcNAc-conjugated PEI complex (1:20) highly interacted with HeLa cells, vimentin-deficient HeLa cells, and NHDFs to the same degree, possibly due to its cationic characteristic.

DLS analysis indicated that AC-GlcNAc-conjugated PEI (1:20) completely condensed NF-κB decoy ODNs, and the size was approximately 50 nm (Figure 4, Table 2). However, the cationic characteristic of AC-GlcNAc-conjugated PEI (1:20) might have been too strong to specifically interact with vimentin-expressing cells. Fluorescent imaging showed similar results to those of flow cytometry (Figure 6). The AC-GlcNAc-conjugated PEI complexes (1:5 and 1:10) were accumulated on HeLa cells and NHDFs, but not on vimentin-deficient HeLa cells. In particular, AC-GlcNAc-conjugated PEI (1:10) was completely condensed and the size was approximately 100 nm. We considered that this complex retained NF-κB decoy ODNs stably. These results demonstrated that AC-GlcNAc-conjugated PEI complexes (1:5 and 1:10) were optimal for targeting vimentin-expressing cells via cell-surface vimentin.

### 3.4. Estimation of Cellular Uptake of AC-GlcNAc-conjugated PEI Complexes Including NF-κB Decoy ODNs and HSP47-siRNA

First, to evaluate cellular uptake and the anti-inflammatory effect of AC-GlcNAc-conjugated PEI complexes, we examined whether the expression of TNF-α in LPS-stimulated NHDFs was suppressed by the cellular uptake of NF-κB decoy ODNs with AC-GlcNAc-conjugated PEI complexes. NHDFs were stimulated with 5 µg/mL LPS, and AC-GlcNAc-conjugated PEI complexes (1:5 and 1:10) were added to these cells. At 8 h after the addition, the expression of TNF-α in each cell type was examined by western blotting. The expression of TNF-α in NHDFs increased with 5 µg/mL LPS stimulation, whereas the upregulation of TNF-α in 5 µg/mL LPS-stimulated NHDFs was suppressed by the addition of AC-GlcNAc-conjugated PEI complexes (1:5 and 1:10) (Figure 7a). Second, we examined the knockdown of HSP47 in TGF-β-stimulated NHDFs by the cellular uptake of HSP47-siRNA with AC-GlcNAc-conjugated PEI (1:10). The upregulation of HSP47 by adding TGF-β1 was observed and the upregulation of HSP47 in TGF-β-stimulated NHDFs was inhibited to same level of control (TGF-β-unstimulated NHDFs) by the cellular uptake of HSP47-siRNA with AC-GlcNAc-conjugated PEI (1:10) (Figure 7b).

Since the expression of β-actin as a housekeeping gene was not decreased in AC-GlcNAc-conjugated PEI complex-treated NHDFs, the cytotoxicity of these complexes was estimated to be low (Figure 7). Moreover, the cytotoxicity of AC-GlcNAc-conjugated PEI complexes toward NHDFs was evaluated by CCK-8 assays. The cell viability of NHDFs treated with AC-GlcNAc-conjugated PEI complexes was approximately >85% (Appendix A). Therefore, the cytotoxicity of PEI and these complexes was indicated to be low. These results demonstrate that AC-GlcNAc-conjugated PEI complexes are effectively taken up into the cytoplasm by cell-surface vimentin without the cytotoxicity (Figure 8).

## 4. Discussion

AC-GlcNAc-conjugated PEI was designed as the gene carrier that can specifically target cells expressing vimentin on their surface. It is composed of a GlcNAc-bearing polymer as the vimentin-targeting moiety and PEI as the gene-carrying moiety, and these were linked by disulfide bridges. We succeeded in producing AC-GlcNAc regulated at approximately 13 GlcNAc ligands and having the carboxyl and thiol group at both terminal sides via RDRP. The sizes and components of AC-GlcNAc-conjugated PEI were reproducibly produced due to the well-defined AC-GlcNAc. Therefore, the size and cationic characteristic of complexes can be controlled by altering the contents of each moiety. In this study, the optimal ratios of AC-GlcNAc to PEI to target cell-surface vimentin-expressing cells were found to be 1:5 and 1:10. The size of the AC-GlcNAc-conjugated PEI complex (1:10) was approximately 100 nm, and this size might be optimal for cellular uptake. These complexes were taken up into the cytoplasm through cell-surface vimentin-mediated endocytosis and were then probably incorporated into endosomes and lysosomes [21,22,23]. These complexes (i.e., the conjugation of AC-GlcNAc to PEI and the self-assembly of PEI) were formed by disulfide bonds. Therefore, these complexes are likely disassembled under intracellular reductive conditions [17], releasing NF-κB decoy ODNs and HSP47-siRNA from endosomes by the “proton sponge” effect of PEI [39] (Figure 8). Moreover, since the molecular weight of PEIs that formed the complexes was below 2 kDa, the cytotoxicity of these PEIs released from the complexes is expected to be low [40].

Currently, there are reportedly no effective therapeutic approaches for fibrosis except anti-inflammatory treatments as palliative therapy. In potent therapeutic approaches for various fibroses, it is important to target myofibroblasts and activated stellate cells in lesion sites. However, it is difficult to target myofibroblasts and activated stellate cells because there are no specific cell surface-markers to target these cells. Many studies reported that platelet-derived growth factor-β receptor, translocator protein, and integrin α_v_β_5_ and α_v_β_3_, as biomarkers of fibroblasts and stellate cells, are useful to target myofibroblasts and activated stellate cells [41]. However, we assume that these molecules might not be rather specific for the lesion sites of fibrosis because of their expression in normal tissues. Cell-surface vimentin and desmin could be specifically expressed on myofibroblasts and activated stellate cells [22,23]. Therefore, our therapeutic approach would be advantageous for applications in clinical settings for various fibroses.

AC-GlcNAc-conjugated PEI can engage various ODNs and siRNA. It can be also used for various target cell-surface vimentin-expressing cells, such as malignant tumor cells. Previously, many kinds of tumor cells have been reported to express vimentin on their surface [42,43,44,45]. Therefore, AC-GlcNAc-conjugated PEI could be advantageous for the delivery of siRNAs that have tumor-suppressive effects, such as survivin siRNA, to these tumor cells.

## 5. Conclusions

In this study, we succeeded in designing a gene carrier AC-GlcNAc-conjugated PEI to specifically target cell-surface vimentin-expressing cells such as myofibroblasts and activated stellate cells by controlling the binding amount of PEI to AC-GlcNAc. Since thiol group-added and well-defined AC-GlcNAc was precisely synthesized via RDRP, the component of AC-GlcNAc-conjugated PEI was reproducibly produced. The regulation of the binding amount of PEI to AC-GlcNAc could create an optimal gene carrier that possesses the ability to strongly retain ODNs and specifically recognize cell-surface vimentin. AC-GlcNAc-conjugated PEI/NF-κB decoy ODNs and HSP47-siRNA complexes were demonstrated to be specifically taken up into NHDFs via cell-surface vimentin. Moreover, AC-GlcNAc-conjugated PEI that can carry various ODNs and siRNA would be an advantageous therapeutic approach for various fibroses and tumors.

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
