# Peer review of "Development of a Gene Delivery System of Oligonucleotides for Fibroses by Targeting Cell-Surface Vimentin-Expressing Cells with N-Acetylglucosamine-Bearing Polymer-Conjugated Polyethyleneimine"

_polymers, 2020, doi:10.3390/polym12071508_

Round 1
Reviewer 1 Report
Comments:
This paper from Ise and coworkers synthesized a N-acetylglucosamine-bearing polymer-conjugated polyethyleneimine by radical polymerization, as the gene carrier to target cell-surface vimentin- expressing cells and specifically deliver nuclear factor-κB decoy oligonucleotides (ODNs) to normal human dermal fibroblasts (NHDFs) that express cell-surface vimentin. The results showed that effective and specific delivery of ODNs and small interfering RNAs for cell-surface vimentin-expressing cells such as myofibroblasts and activated stellate cells can be achieved using GlcNAc-bearing polymer-conjugated PEI. This therapeutic approach could prove advantageous for preventing the promotion of various fibroses. The most of results and discussion are reasonable. Indeed the work is interesting, the paper is well prepared. The manuscript would be recommended for publication in Polymers.
Please improve the clear 1H NMR spectrum of polymer.
Author Response
We appreciate for reviewing our manuscript titled " Development of gene delivery system for oligonucleotides to lesion sites of fibrotic tissues by N-acetylglucosamine-bearing polymer-conjugated polyethyleneimine" (polymers-765603). We described our responses to the reviewer’s comments.
The reviewer’ comment 1
Please improve the clear 1H NMR spectrum of polymer.
Response
Thank you very much for your comment. According to the reviewer’s suggestion, we added the larger 1H NMR spectrum of polymer to Figures 1, 2, and S1 (supplementary materials).
Reviewer 2 Report
The work is interesting and practical which is potentially helpful for those to investigate the Development of gene delivery system. Overall, authors have done a good job in providing results from their studies and discussion. I would recommend author of this paper to please address following comment:
- Conclusion section of this paper is missing. Could author clearly mention what are the conclusion from this study.
Author Response
We appreciate for reviewing our manuscript titled " Development of gene delivery system for oligonucleotides to lesion sites of fibrotic tissues by N-acetylglucosamine-bearing polymer-conjugated polyethyleneimine" (polymers-765603). We described our responses to the reviewer’s comments.
The reviewer’ comment 2
Conclusion section of this paper is missing. Could author clearly mention what are the conclusion from this study.
Response
Thank you very much for your comment. According to the reviewer’s suggestion, we added the following sentences to “5. Conclusion”.
“In this study, we succeeded in designing a gene carrier AC-GlcNAc-conjugated PEI to specifically target cell-surface vimentin-expressing cells such as myofibroblasts and activated stellate cells by controlling the binding amount of PEI to AC-GlcNAc. The regulation of the binding amount of PEI to AC-GlcNAc could create the optimal gene carrier that possesses the ability to strongly retain ODNs and specifically recognize cell-surface vimentin. AC-GlcNAc-conjugated PEI and NF-kB decoy ODN complexes were demonstrated to be specifically taken up into HeLa cells and NHDFs by cell-surface vimentin. Moreover, AC-GlcNAc-conjugated PEI that can carry various ODNs such as siRNA would be an advantageous therapeutic approach for various fibroses and tumors.”
Reviewer 3 Report
In this work, Song and Ise report the synthesis of N-acetylglucosamine-conjugated polyethyleneimine (PEI) and initial in-vitro study on the interaction between the conjugated PEI with HeLa cells. The chemistry is certainly not new as the authors employed a well-know Micheal addition. The use of a toxic polycation PEI for a biological application is unjustified and not reasonable. The motivation of the work is to “precisely control the degree of GlcNAc-conjugation”. However, the polymer synthesized is not well-characterized, and the goal of the work is not achieved. The authors may need assistance from a chemist to present NMR spectra (Figure 1 and Figure 2) with correct phasing (phase correction). Furthermore, the use of a cancer cell line (HeLa) cannot justify the aim stated in the title “Development of gene delivery system for oligonucleotides to lesion sites of fibrotic tissues”. Important, no knockdown efficiency was reported, so the purpose of siRNA delivery has not been achieved. Overall the work is not novel and not completed. The manuscript does not yet make any advance to the field of gene delivery and therefore publishing this manuscript as this stage is premature.
Author Response
We appreciate for reviewing our manuscript titled " Development of gene delivery system for oligonucleotides to lesion sites of fibrotic tissues by N-acetylglucosamine-bearing polymer-conjugated polyethyleneimine" (polymers-765603). We described our responses to the reviewer’s comments.
Reviewer’ comment
In this work, Song and Ise report the synthesis of N-acetylglucosamine-conjugated polyethyleneimine (PEI) and initial in-vitro study on the interaction between the conjugated PEI with HeLa cells. The chemistry is certainly not new as the authors employed a well-know Micheal addition. The use of a toxic polycation PEI for a biological application is unjustified and not reasonable. The motivation of the work is to “precisely control the degree of GlcNAc-conjugation”. However, the polymer synthesized is not well-characterized, and the goal of the work is not achieved. The authors may need assistance from a chemist to present NMR spectra (Figure 1 and Figure 2) with correct phasing (phase correction). Furthermore, the use of a cancer cell line (HeLa) cannot justify the aim stated in the title “Development of gene delivery system for oligonucleotides to lesion sites of fibrotic tissues”. Important, no knockdown efficiency was reported, so the purpose of siRNA delivery has not been achieved. Overall the work is not novel and not completed. The manuscript does not yet make any advance to the field of gene delivery and therefore publishing this manuscript as this stage is premature.
Response
1.Cytotoxicity of AC-GlcNAc-conjugated PEI (1:1), (1:5), (1:10), and (1:20) complexes against NHDFs was examined by Cell Counting Kit-8 assay. The cell viability of these complexes-treated NHDFs was more than 85%. Therefore, the cytotoxicity of AC-GlcNAc-conjugated PEI was low. We added these data to Figure S3. Moreover, the following sentences were added to Results section.
“Since the expression of b-actin as a housekeeping gene was not decreased in AC-GlcNAc-conjugated PEI complex-treated NHDFs, the cytotoxicity of these complexes was estimated to be low (Figure7). Moreover, the cytotoxicity of AC-GlcNAc-conjugated PEI complexes toward NHDFs was evaluated by CCK-8 assays. The cell viability of NHDFs treated with AC-GlcNAc-conjugated PEI complexes was approximately > 85% (Figure S3). Therefore, the cytotoxicity of PEI and these complexes was indicated to be low.”
- The detailed characterization of AC-GlcNAc was described in the previous report (Ise et al., Elucidation of GlcNAc‐binding properties of type III intermediate filament proteins, using GlcNAc‐bearing polymers. Genes to Cells 2017, 22, 900-917, ref.14). In this study, the characterization of AC-GlcNAc was confirmed by GPC (Figure S1) and NMR. We considered that AC-GlcNAc was precisely synthesized by RAFT polymerization.
Phase correction was performed in NMR spectra of CBA-conjugated PEI and AC-GlcNAc-conjugated PEI. New NMR spectra were added in Figure 1, 2, and S2.
- To design a gene carrier to specifically and effectively target cell-surface vimentin of myofibroblasts and activated stellate cells, the optimal gene carrier that can strongly retain oligonucleotides (ODNs) and specifically recognize cell-surface vimentin without nonspecific cell-interactions induced by the cationic character must be created. The optimal AC-GlcNAc-conjugated PEI was made by regulating the binding ratio between disulfide bridge-introduced PEI and thiol-introduced AC-GlcNAc. HeLa cells (vimentin-expressing cells) and vimentin-deficient HeLa cells were used to determine the optimal binding ratio for specifically interacting with cell-surface vimentin of NHDFs. The complexes that could interact with HeLa cells but not with vimentin-deficient HeLa cells were determined as optimal ones. In this study, NHDFs were used as myofibroblasts and activated stellate cells to examine the gene delivery of NF-kB decoy ODNs because NHDFs that were cultured for long-term express a-smooth muscle actin and have myofibroblast-like characters. The effective gene delivery of NF-kB decoy ODNs can induce anti-inflammatory effects in these cells and would be a useful approach for improving and suppressing tissue fibroses.
We added the following sentence to Introduction section.
“To specifically and effectively target cell-surface vimentin of myofibroblasts and activated stellate cells, nonspecific interactions induced by the cationic character of PEI with cells must be suppressed. AC-GlcNAc-conjugated branched PEI that can interact with HeLa cells but not with vimentin-deficient HeLa cells is presumed to be optimal for the specific recognition of cell-surface vimentin. We determined the optimal conjugation ratio of branched PEI to AC-GlcNAc by altering the amount of PEI conjugated to AC-GlcNAc using HeLa cells and vimentin-deficient HeLa cells. Moreover, normal human dermal fibroblasts (NHDFs) were used as myofibroblasts and activated stellate cells in this study. NHDFs that are cultured long-term express a-smooth muscle actin and have myofibroblast-like characteristics. To develop a gene carrier to target lesion sites of fibrotic tissues, the interaction and the anti-inflammatory effect of NHDFs with AC-GlcNAc-conjugated PEI/NF-kB decoy ODNs complexes were examined.”
- This manuscript suggested a therapeutic approach for various fibroses by using a gene delivery system for ODNs to lesion sites of fibrotic tissues. The effectively therapeutic approach for fibrosis is important to suppress chronic inflammation of fibrotic lesion sites. Therefore, we examined whether the effective delivery of NF-kB decoy ODNs has strong anti-inflammatory effects on NHDFs. Since NF-kB decoy ODNs can inhibit expressing many cytokines regarding inflammation, the delivery of NF-kB decoy ODNs to the lesions would be the more effective improvement and suppression of various fibroses than that of siRNA. Therefore, in this study, the specific delivery of NF-kB decoy ODNs to NHDFs was examined by AC-GlcNAc-conjugated PEI. Since it is expected that AC-GlcNAc-conjugated PEI can also deliver various ODNs such as siRNA, we proposed siRNA delivery by using AC-GlcNAc-conjugated PEI in this manuscript. To avoid the misunderstanding, we described the proposal of siRNA delivery by our approach in the discussion section only. In future work, the delivery of siRNA by AC-GlcNAc-conjugated PEI to cell-surface vimentin expressing cells such as tumor cells will be performed.

Round 2
Reviewer 3 Report
In this revised version, the authors have spent a lot of efforts to significantly improve the quality of the work and the manuscript. Several experiments have been added including cytotoxicity of AC-GlcNAc-conjugated PEI and characterization of polymers by GPC and NMR. However, still no knockdown efficiency is reported, so the purpose of siRNA delivery has not been achieved.
The authors explained in the response that “Since it is expected that AC-GlcNAc-conjugated PEI can also deliver various ODNs such as siRNA, we proposed siRNA delivery by using AC-GlcNAc-conjugated PEI in this manuscript. To avoid the misunderstanding, we described the proposal of siRNA delivery by our approach in the discussion section only. In future work, the delivery of siRNA by AC-GlcNAc-conjugated PEI to cell-surface vimentin expressing cells such as tumor cells will be performed. ” This explanation is conflicted with what written in the manuscript. The title is unchanged and gene delivery is still the key focus of the work presenting in the title “Development of gene delivery system for oligonucleotides to lesion sites of fibrotic tissues…”. In the introduction, the authors still discuss the literature and the motivation of the work in gene delivery “the production of gene carriers that are conjugated to these GlcNAc-bearing polymers might be advantageous for targeting myofibroblasts and activated stellate cells…” In the conclusion, the authors still stated that “we succeeded in designing a gene carrier…”
Overall, this work clearly focuses on gene delivery and therefore, I recommend the authors perform real gene knockdown study. It would be incremental to do this experiment in the future work. And a gene knockdown study is not difficult and not time-consuming. Another minor note, the title should change “lesion sites of fibrotic tissues” to “vimentin-expressing HeLa cells” because this work does not have data showing the delivery to lesion sites of fibrotic tissues, which can only be demonstrated by in-vivo experiment.
Author Response
We appreciate for reviewing our manuscript titled " Development of gene delivery system for oligonucleotides to lesion sites of fibrotic tissues by N-acetylglucosamine-bearing polymer-conjugated polyethyleneimine" (polymers-765603). We described our responses to the reviewer’s comments.
Reviewer’ comment
In this revised version, the authors have spent a lot of efforts to significantly improve the quality of the work and the manuscript. Several experiments have been added including cytotoxicity of AC-GlcNAc-conjugated PEI and characterization of polymers by GPC and NMR. However, still no knockdown efficiency is reported, so the purpose of siRNA delivery has not been achieved.
The authors explained in the response that “Since it is expected that AC-GlcNAc-conjugated PEI can also deliver various ODNs such as siRNA, we proposed siRNA delivery by using AC-GlcNAc-conjugated PEI in this manuscript. To avoid the misunderstanding, we described the proposal of siRNA delivery by our approach in the discussion section only. In future work, the delivery of siRNA by AC-GlcNAc-conjugated PEI to cell-surface vimentin expressing cells such as tumor cells will be performed. ” This explanation is conflicted with what written in the manuscript. The title is unchanged and gene delivery is still the key focus of the work presenting in the title “Development of gene delivery system for oligonucleotides to lesion sites of fibrotic tissues…”. In the introduction, the authors still discuss the literature and the motivation of the work in gene delivery “the production of gene carriers that are conjugated to these GlcNAc-bearing polymers might be advantageous for targeting myofibroblasts and activated stellate cells…” In the conclusion, the authors still stated that “we succeeded in designing a gene carrier…”
Overall, this work clearly focuses on gene delivery and therefore, I recommend the authors perform real gene knockdown study. It would be incremental to do this experiment in the future work. And a gene knockdown study is not difficult and not time-consuming. Another minor note, the title should change “lesion sites of fibrotic tissues” to “vimentin-expressing HeLa cells” because this work does not have data showing the delivery to lesion sites of fibrotic tissues, which can only be demonstrated by in-vivo experiment.
Response
- According to the reviewer’s suggestions, we performed the knockdown experiment by using AC-GlcNAc-conjugated PEI/siRNA complexes. In TGF-b -stimulated NHDFs, the knockdown of HSP47 by AC-GlcNAc-conjugated PEI/HSP47-siRNA was examined. The upregulation of HSP47 by TGF-b1 was observed and the upregulation was inhibited by AC-GlcNAc-conjugated PEI/HSP47-siRNA complexes. These results indicated that HSP47-siRNA could be incorporated into the cytoplasm by AC-GlcNAc-conjugated PEI complexes. These results are added to the Results section, Figure 7b, and Figure 8.
- According to the reviewer’s suggestions, the title was rewritten to “Development of gene delivery system of oligonucleotides for fibroses by targeting cell-surface vimentin-expressing cells with N-acetylglucosamine-bearing polymer-conjugated polyethyleneimine”.

Round 3
Reviewer 3 Report
Thanks to the authors' efforts to improve the work, the manuscript is now recommended to publish on Polymers.